# Predicting rhizosphere-competence-related catabolic gene clusters in plant-associated bacteria with rhizoSMASH

Yuze Li [1,2,3,4], Mingxue Sun [1,2,3,4], Jos M. Raaijmakers [4,5], Liesje Mommer [6], Fusuo Zhang [1,2], Chunxu Song [1,2] ✉ & Marnix H. Medema [3,5] ✉

Plants release a substantial fraction of their photosynthesized carbon into the rhizosphere as root exudates that drive microbiome assembly. Deciphering how plants modulate the composition and activities of rhizosphere microbiota through root exudates is challenging, as no dedicated computational methods exist to systematically identify microbial root exudate catabolic pathways. Here, we integrate published information on catabolic genes in bacteria that contribute to their rhizosphere competence and develop the rhizoSMASH algorithm for genome-synteny-based annotation of rhizosphere-competence-related catabolic gene clusters (rCGCs) in bacteria with 58 curated detection rules. Our analysis reveals heterogeneity in rCGC prevalence both across and within plant-associated bacterial taxa, indicating extensive niche specialization. Furthermore, we demonstrate the predictive value of the presence or absence of rCGCs for rhizosphere competence in machine learning with two case studies. rhizoSMASH provides an extensible framework for studying rhizosphere bacterial catabolism, facilitating microbiome-assisted breeding approaches for sustainable agriculture.

The rhizosphere, the narrow soil zone surrounding and influenced by plant roots, exhibits physicochemical properties distinct from bulk soil, shaped by root-mediated carbon deposition and metabolic activity. Plants release a significant amount of photosynthesized carbohydrates into their rhizosphere[1,2]. These exudates comprise a complex blend of chemical compounds, including sugars, organic acids, amino acids, biogenic amines, and secondary metabolites, which can vary substantially across plant species, genotypes within a plant species, developmental stages, and physiological conditions[3–5]. Acting as nutrients, chemoattractants, and antimicrobials, root exudates shape the composition and activities of microbial communities in the rhizosphere[4,6,7]. Bacteria adapted to colonizing this niche, collectively

termed rhizobacteria, form part of the plant beneficial microbiome and can enhance nutrient acquisition, regulate phytohormone homeostasis, and mitigate (a) biotic stresses[8–18].

Given the vast and dynamic chemical diversity of root exudates, deciphering how plants metabolically modulate rhizosphere community composition and activity remains a major challenge[19]. Understanding rhizobacterial catabolic pathways is crucial for predicting host selection of their microbiota and the subsequent rhizobacterial colonization success[20–24]. Some studies have shown that the dysfunction of a single catabolic pathway can directly affect rhizosphere competence, i.e., the ability of microbial taxa to colonize the rhizosphere in competition with other microorganisms[25–28]. Thus, linking

[1]State Key Laboratory of Nutrient Use and Management, College of Resources and Environmental Sciences, National Academy of Agriculture Green Development, Key Laboratory of Plant-Soil Interactions, Ministry of Education, China Agricultural University, Beijing, China. [2]National Observation and Research Station of Agriculture Green Development, Quzhou, Hebei, China. [3]Bioinformatics Group, Wageningen University, Wageningen, The Netherlands. [4]Microbial Ecology Department, NIOO-KNAW, Wageningen, The Netherlands. [5]Institute of Biology, Leiden University, Leiden, The Netherlands. [6]Forest Ecology and Forest Management Group, Wageningen University, Wageningen, The Netherlands. ✉e-mail: chunxu.song@cau.edu.cn; marnix.medema@wur.nl

bacterial catabolic capacities to plant root exudate profiles is instrumental for predicting rhizosphere competence.

Predicting bacterial catabolic capacities from genomic data remains challenging due to functional diversification within enzyme families. Homology-based approaches often fail to accurately annotate enzyme-coding genes, as evolutionarily divergent enzymes within shared protein families frequently exhibit high sequence similarity yet catalyze distinct reactions. For example, in *Pseudomonas* spp. genomic annotations, genes encoding L-lysine mono-oxygenases tend to be misannotated as L-tryptophan monooxygenase, because they are from the same protein family (PF01593) and have high mutual sequence similarity[15,29,30]. The annotation accuracy of metabolic genes can be improved by leveraging their genomic synteny context[31]. Genes encoding enzymes from a catabolic pathway often co-localize in the genome, forming operons or larger gene clusters. This arrangement not only facilitates coordinated gene expression in polycistronic mRNAs, but also enables related genes being translocated together as a whole mobile unit[32,33]. This phenomenon enables distinguishing the above-mentioned L-lysine and L-tryptophan-monooxygenase-coding genes based on whether they cluster with a carbon-nitrogen hydrolase family (PF00795) or an amidase family (PF01425) gene[15]. The ability to accurately and systematically identify rhizosphere-competence-related catabolic gene clusters (rCGCs) would facilitate establishing functional links between plant metabolic diversity and their microbiome composition and thus enable breeding strategies to steer plant microbiome composition through specific constituents in the root or shoot exudates.

Here, we leveraged synteny-based annotation principles to address this challenge, and developed rhizoSMASH (**rhizo**sphere-competence-related catabo**liSM A**nalysis **SH**ell), a bioinformatic tool which applies a rule-based gene cluster detection algorithm based on the successful principles developed in antiSMASH[34] to facilitate the prediction of gene clusters related to the catabolism of root exudate metabolites. The first distribution of rhizoSMASH contains over 50 rCGC detection rules, carefully manually curated based on sequence similarity network analysis and covering a wide range of gene clusters encoding pathways that catabolize specific carbohydrates, organic acids, amino acids, biogenic amines, phytohormones, and aromatic compounds found in root exudates. We screened soil- and rhizo-bacterial genomes with rhizoSMASH and revealed patterns in taxonomic and genomic distribution of rCGCs. Our analyses on two case

studies also verified the connection between catabolic capacities and the genomic presence of rCGCs, indicating that the predictive value of rhizoSMASH-based rCGC absence/presence profiles on rhizosphere competence is 'on par' with that of experimentally measured substrate utilization assays.

## Results and discussion
### Development of rhizoSMASH
We introduced a first working version of the **rhizoSMASH** algorithm that predicts rCGC in bacterial genomes. As a member of the antiSMASH software family[34,35], rhizoSMASH also predicts gene clusters using a set of detection rules. Each of these rules describes the combination(s) of functional domains captured by profile hidden Markov models (Fig. 1a, for more details see and *Methods*).

The development of rhizoSMASH started with an initial set of detection rules summarized from known rCGCs with genetic, and/or biochemical evidence for their function in literature (Supplementary Fig. 1, Supplementary Table 2). The detection rules then underwent three rounds of manual calibration using sequence similarity network analysis to improve their prediction accuracies based on detailed literature-guided assessment of detected gene cluster families to identify putative false positives and false negatives (Fig. 1b, final verification results are available at https://www.bioinformatics.nl/-li286/rhizosmash-demo, and more details are in *Methods*)[36]. The final set consists of 58 detection rules, covering pathways that catabolize six chemical classes of substrates commonly found in root exudates, which are carbohydrates, organic acids, amino acids, amines, phytohormones and aromatic compounds. The explanations of these substrates, their catabolism in bacteria, examples of known rCGCs, and evidence for their relevance in rhizosphere competence are documented at https://www.bioinformatics.nl/-li286/rhizosmash-doc.

In the working version of rhizoSMASH, we largely focused on catabolic pathways of soluble primary metabolites (with several exceptions). However, macromolecules, plant-derived polysaccharides, volatile compounds, and plant-species-specific secondary metabolites have also been shown to affect rhizosphere bacteria-plant interactions[2,7,37–41]. RhizoSMASH is an extendable software similar to the other members from the antiSMASH family, and more detection rule covering catabolism pathways utilizing the above-mentioned metabolite repertoires will be added into rhizoSMASH in the future versions.

### Diversity of rCGC profile in soil and rhizosphere bacteria
We constructed a collection of 1,226 genome assemblies from **BA**cteria in the **R**hizosphere and **S**oil (the **BARS** collection, see details in *Methods* and Supplementary Table 1) to study the distribution of rCGCs across bacterial taxonomy. These genomes exhibited both between- and within-clade diversity in their rCGC presence/absence profiles (Supplementary Fig. 2). We summarized the prevalence of each rCGC type in 20 bacterial families with at least 8 complete assembled genomes in the BARS collection (Fig. 2a). Our results showed that several rCGC types have high prevalence across almost all bacterial families (e.g., glutamate synthase *glt* cluster, 79.3% and glutamine synthetase *gln* cluster, 93.6%), indicating that these gene clusters may encode pathways of fundamental roles in catabolism that are present in most rhizobacterial genomes. Other rCGC types are specific to a limited range of clades: the L-proline catabolizing *put* clusters are almost only found in *Pseudomonadota* (previously known as *Proteobacteria*) and a few *Bacillota* (previously known as *Firmicutes*) genomes, and the D-proline racemase pathway *prd* clusters are almost limited to *Clostridiaceae*[42]. Some gene cluster types encoding pathways with the same substrate, such as the sucrose hydrolase, phosphorylase, and the levansucrase detour pathways, share overlapping distributions across taxonomy. However, for some other cases, such as the trehalose phosphotransferase and the trehalase pathways, they

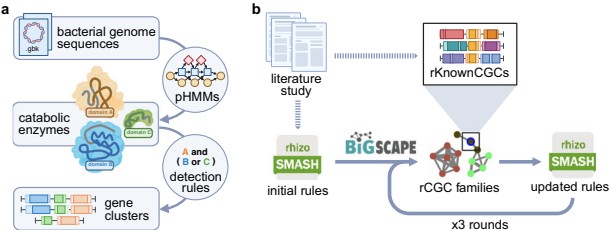

**Fig. 1 | Development of rhizoSMASH. a** The gene cluster prediction workflow of rhizoSMASH. RhizoSMASH takes a genome sequence file as input (GenBank or FASTA) and recognize potential catabolic enzymes by scanning the sequence profile hidden Markov models. Gene clusters encoding relevant pathways were then detected using a set of detection rules. **b** The tuning procedure used for curation of rCGC detection rules. An initial set of detection rules was first summarized from a comprehensive literature study. Then, genome sequences in our BARS collection were scanned using this set of detection rules. The output gene clusters were grouped into cluster families with BiG-SCAPE together with our known cluster database, rKnownCGCs. We manually curated the detection rules by visually investigating the gene cluster family network generated by BiG-SCAPE for putative false positives/negatives, aided by further literature searches when needed. This calibration, validation and finetuning was performed three times to arrive at more and more optimal detection rules. Created with BioRender.com.

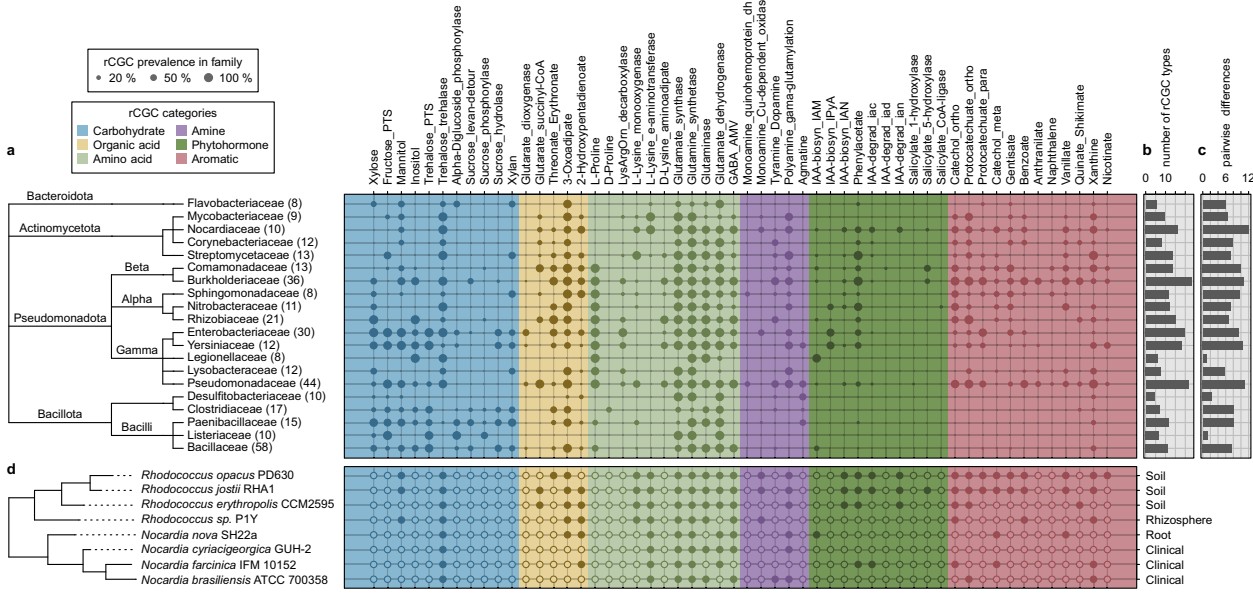

**Fig. 2 | Distribution of rCGCs across bacterial taxonomy. a** rCGC prevalence in bacterial families. The size of a dot represents the percentage of genomes in a family that an rCGC type has been detected by rhizoSMASH (defined as prevalence in the main text). The phylogenetic tree on the left was derived from the lineage information available at NCBI. The number in parentheses indicates the number of completely assembled genome in BARS for each family. **b** The richness (the average number of rCGC types present in the genomes of a family) and **c** the diversity (the average differences in rCGC presence/absence between each pair of genomes in a family) indexes of rCGC types across bacterial families. **d** The rCGC presence/absence profiles in *Nocardiaceae* spp. genomes whose isolation origin were clear, where a solid dot represents the presence of a rCGC type in a genome, and an open dot represents the absence. The labels on the right represent the isolation origins, based on their records at JGI GOLD and NCBI. The phylogeny on the left was constructed with genomic 16S rDNA sequences, rooted with the *Psedomonas putida* type strain NBRC 14164 as an outgroup (not shown on the tree).

display complementary distribution, suggesting niche differentiation across different taxa (*phi* = −0.27, Fisher's exact, *P* = 4 × 10⁻⁷). Note that the absence of an rCGC in a genome does not necessarily mean the genome does not carry any genes encoding the pathway; in rare cases, these genes may be scattered in the genome and, therefore, not recognized as a gene cluster in rhizoSMASH. Also, because the rCGC detection rules were designed based on known studies, catabolic pathways encoded by undocumented gene cluster may also exist.

Non-metric multidimensional scaling (NMDS) of the prevalence of the rCGC of 20 bacterial families (Supplementary Fig. 3) showed taxonomically related families to share similar rCGC repertoires (PERMANOVA of the null: familywise rCGC prevalences are the same across all phyla, *P* = 0.001). Specifically, *Bacillota* genomes contain more carbohydrate-catabolism-associated rCGCs (Wilcoxon, *P* = 0.0219; *Bacillota* sectors in Fig. 2a and Supplementary Fig. 3), consistent with a previous phylogenetic study on the distribution of carbohydrate-activate enzymes in bacteria[43]. In contrast, aromatic-compound- and phytohormone-catabolizing rCGCs are more frequently found in *Pseudomonadota* and *Actinomycetota* (Wilcoxon, *P* < 2.2 × 10⁻¹⁶; Corresponding sectors in Fig. 2a and Supplementary Fig. 2). Indeed, most known bacterial aromatic catabolic pathways[44,45], as well as auxin biosynthesis and degradation pathways[46,47], were characterized within these two phyla.

Subsequently, we want to zoom from a broader taxonomic rank into each bacterial family. To this end, we first investigated the richness and diversity (detailed definitions in *Methods: Family-wise distribution of rCGCs*) of rCGC types in each family (Fig. 2b, c, Supplementary Fig. 2). The results showed that families with the highest richness of rCGCs generally belong to *Pseudomonadota*. Within *Pseudomonadota*, the families *Burkholderiaceae* and *Pseudomonadaceae* have the highest richness and diversity in rCGC types in their genomes, indicating that many of these organisms are metabolic generalists able to grow on a wide array of root exudate components. These two families are also

frequently found as members of the most dominant bacteria families in various rhizosphere microbiomes[48,49]. Especially, the family *Pseudomonadaceae* harbors a large number of known plant growth-promoting rhizobacterial strains[9,50–56]. In contrast, genomes in the families *Desulfitobacteriaceae*, *Listeriaceae* and *Legionellaceae* displayed both the lowest richness and diversity in rCGC types (Fig. 2b, c, Supplementary Fig. 2). These families are generally not enriched in the rhizosphere: *Desulfitobacteriaceae* spp. bacteria are strict anaerobic sulfate or organohalide reducers commonly found in anoxic sediment[57,58]; *Listeriaceae* and *Legionellaceae* are families whose members are known as food-borne or environmental pathogens[59,60]. All genome accessions from these three families belong to the RefSoil subcollection, indicating their presence in various soil samples based on metagenomic reads that were mapped to their 16S rDNA sequences[61], but a detailed inspection of these strains showed that none of them were originally isolated from plant-associated environments including the rhizosphere (Supplementary Table 3). These results suggest that rhizosphere-dwelling bacteria carry more abundant and diverse rCGCs than bacteria adapted to ecosystems other than plants.

Subsequent focus on the family *Nocardiaceae*, which showed the highest diversity of rCGC types across all tested families (Fig. 2a, d), suggested that members of this family may have different catabolic strategies to adapt to their local environments. *Nocardiaceae* strains in our study were originally isolated from various ecosystems, including human patients, soil, rhizosphere and root (Supplementary Table 3) which was reflected in their phylogeny (Fig. 2d). In general, environmental strains tend to carry more rCGC types compared to clinical *Nocardiaceae* strains (Wilcoxon, *P* = 0.018). Soil-derived strains are abundant in aromatic compound degradation pathways (Fig. 2d), which enable them to utilize lignin-derived materials that are abundant in soil[62,63]. The existence of auxin-metabolizing pathways in *Nocardiaceae* spp. has been reported in several studies[17,64–68]. RhizoSMASH

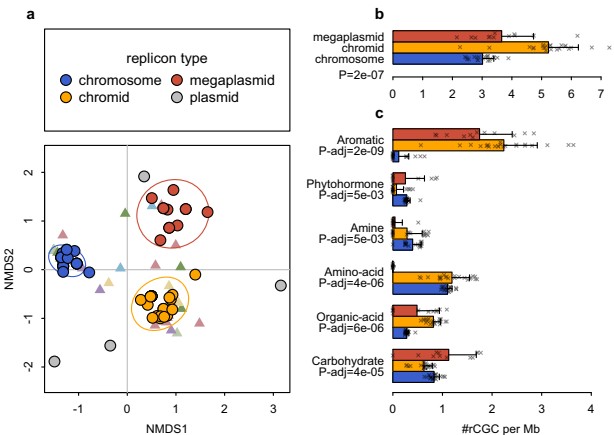

**Fig. 3 | Biased distribution of rCGC in *Burkholderia* spp. replicons. a** NMDS plot of *Burkholderia* spp. replicons according to the presence/absence of rCGCs in each replicon sequences. Each point represents a replicon in a strain, while the color of the dot represents the type of replicon (manually labeled according to their sequence lengths). The ellipse represents the 1-SD range for each type of replicon. Each triangle marker represents a rCGC type, colored according to the same schema used in Fig. 2. **b** The average number of rCGCs present per million bases in each replicon type (lengths of error bar = 1-SD). The *P*-value was from the result of a Kruskal-Wallis rank sum test on the equality of three replicon types. **c** Similar to (**b**), but the rCGC types were divided into six categories based on their substrate properties. The *P*-values were adjusted for multiple testing correction with the Bonferroni method.

also predicted a group of putative rCGCs encoding both auxin biosynthesis and degradation pathways that were enriched in environmental strains (Fig. 2d), including the phenylacetate degradation *paa* cluster[68] and the indole-3-acetate (IAA) to catechol *iac* gene cluster[17]. The aldoxime dehydratase gene clusters (detected by the IAA biosynthesis IAN pathway rule) were also reported to have certain level of substrate specificity for indole-3-aldoxime (IAOx) in other *Rhodococcus* strains[65,66]. This intra-family diversity of rCGCs suggests roles of related catabolism in local adaptation to different environments.

Zooming further into the sub-genomic level, we also observed unevenly distributed localization of rCGCs within various bacterial genomes (Supplementary Fig. 4). They form sub-chromosomal regions that were enriched with rCGCs, including cases where rCGCs encoding pathways down- or upstream of each other (e.g., a benzoic acid catabolic gene cluster and the downstream catechol 2,3-cleavage gene cluster), suggesting they may form genomic islands (GI) which can facilitate rapid niche adaptation through concerted horizontal gene transfer[69], though we did not have clear statistical evidence supporting the relationship between these rCGC-rich regions and tRNA genes (one of the GI markers, Supplementary Fig. 4). To further study this intra-genomic diversity of rCGC, we focused on the genus *Burkholderia*, which typically possesses a multipartite genome containing three large circular replicons[70,71]. The nomenclature of these replicons varies in different study fields, and in our study, we classified them into three types, "chromosome", "chromid" and "megaplasmid", according to their sequence length in descending order in each genome. We analyzed all the replicons from completely assembled *Burkholderia* genomes in BARS with NMDS using the rCGC presence/absence profile in each replicon sequence (Fig. 3a). The result showed that chromosomes, chromids and megaplasmids from different genomes were grouped together according to their types instead of their source genomes, indicating nonrandom localization of certain rCGC classes within the same replicon types (Fig. 3a). Permutational analysis of variances also confirmed that rCGC presence/absence profiles are significantly different among chromosome, chromid and megaplasmid

(PERMANOVA; $P = 0.001$) but not among genome sources ($P = 1.00$) or species ($P = 0.985$). Meanwhile, rCGCs with different substrate categories also show significant preferences in one or two replicons (Fig 3b, c). For example, aromatic compound rCGCs being enriched in megaplasmids and chromids compared to chromosomes, consistent with replicon functional biases reported in previous studies[71,72]. As secondary replicons (chromid and megaplasmids) have faster evolutionary rates[73], the enrichment of rCGCs in chromid and megaplasmids indicates their importance in environment adaptation[70].

So far, we have analyzed the diversity of rCGCs across different taxonomic and sub-genomic levels. Our results indicate that variation in rCGC profiles of bacterial genomes is related to their lifestyles. Therefore, we hypothesize that the rhizosphere competence of a bacterium can be predicted by its genomic rCGC presence/absence profile. To explore this hypothesis, we conducted two case studies.

### Predicting rhizosphere competence with genomic rCGC profiles

Many studies have aimed to predict rhizosphere competence with various approaches and features[4,49,74,75]. Metabolome analyses[4], whole-genome functional annotations[49,74,75] or experimental studies with synthetic communities[76,77] have paved the way to predict rhizosphere behaviors of bacterial groups and have shown clear host-specific connections with specific catabolic activities or genes. However, most of these approaches are quite laborious and do not yet establish direct functional links between root exudate composition and the associated microbial catabolic pathway genes. Here, we use two of these studies[4,20] to demonstrate the value of rhizoSMASH for advancing rhizosphere competence analysis based on rCGC profiles.

For our first case study, we adopted the catabolic capability data (measured with Biolog phenotype microarray) of 60 phenazine-producing *Pseudomonas* spp. strains and their rhizosphere colonization data in *Arabidopsis thaliana* and potato rhizospheres published by Zboralski et al. in 2020 (more details in *Methods*). The NMDS according to both genomic rCGC profiles and assay-measured catabolic capabilities show separation of strains with various rhizosphere colonization levels (Fig. 4a). PERMANOVA also confirmed that both rCGCs profiles and catabolic capabilities were significantly different between high and low ($P_{rCGC} = 0.001$, $P_{cat} = 0.001$), and between medium and low ($P_{rCGC} = 0.001$, $P_{cat} = 0.001$) rhizosphere colonizers, but not significantly different between high and medium colonizers ($P_{rCGC} = 0.354$, $P_{cat} = 0.061$). Therefore, we merged the rhizosphere competence levels "high" and "medium" into one level "med-high" in the subsequent analysis. Then, we trained random forest models to predict colonization levels in *A. thaliana* and potato rhizospheres with either catabolic capabilities or rCGC profiles as predictors (hereafter referred to as catabolism-based models and rCGC-based models respectively). According to an 8-fold nested cross-validation, for rhizosphere colonization in *A. thaliana*, prediction accuracy of the rCGC-based model was 0.884 (SD = 0.106), while the accuracy of the catabolism-based model was 0.848 (SD = 0.115) (Fig. 4c). For rhizosphere colonization in potato, prediction accuracies of the rCGC-based model and the catabolism-based model were 0.783 (SD = 0.122) and 0.681 (SD = 0.110) (Fig. 4d). Models trained on rCGC profiles were in general at least as accurate as models trained on experimentally measured catabolism capabilities (Fig. 4cd).

For our second case study, we broadened the taxonomic scope of bacteria using the dataset published by Zhalnina et al. in 2018, which contains the rhizosphere competence data in *Avena barbata* for a smaller collection of 39 bacterial isolates but from a wider range of phyla compared to the first case study. This dataset also contains the root exudate metabolite uptake profiles in a subset of 16 isolates measured using exometabolite profiling with 2 to 4 valid repetitions per isolate (more details in *Methods*). For this more diverse set of isolates, the NMDS plots did not show obvious separation of

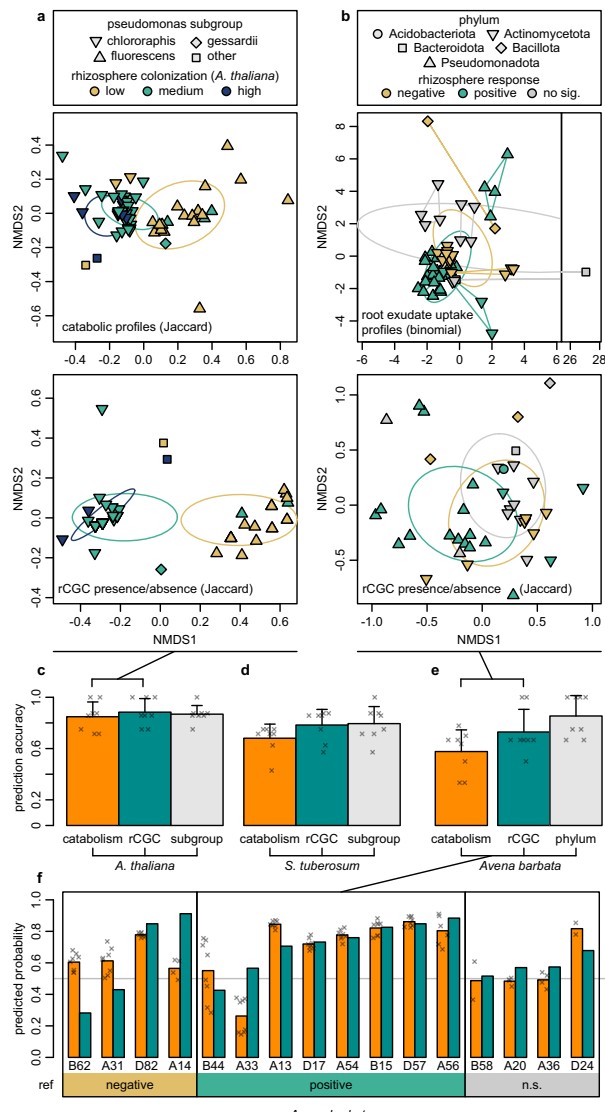

**Fig. 4 | Predicting rhizosphere competence with rCGC profiles. a** NMDS plots of bacterial strains according to their catabolism assay results or genomic rCGC presence/absence profiles in the first case study (*Pseudomonas* spp. isolates) and **b** in the second case study (soil bacterial isolates from Mediterranean grassland). Each point represents a bacterial strain, where the color indicates rhizosphere competence and the shape indicates its *Pseudomonas* subgroup in (**a**) and Phylum in (**b**). Repeated measurements from the same isolate are represented by lines linking datapoints in (**b**). **c–e** Accuracies estimated with cross-validations for prediction models constructed with different datasets. Bar heights represent the mean accuracies derived from 8 cross-validation folds. Cross signs indicate the accuracies estimated in each fold (length of error bar = 1-SD). Labels on the bottom indicates the types of predictors and plant rhizospheres. Note that the catabolism-based model in the Mediterranean grassland dataset (the orange bar in the right group) was evaluated using a different cross-validation setting, because the exometabolite profiling dataset contains repeated observations but on a smaller number of isolates. **f** Predicted class probabilities for 16 exometabolite-profiled isolates in the *Avena barbata* case study. Bar heights represent the out-of-sample predicted probability for positive rhizosphere competence (turquoise: rCGC-based model, orange: catabolism-based model). Cross signs represent predicted probabilities for each repeated measurement.

rhizosphere competence levels according to either rCGC profiles or root exudate metabolite uptake data (Fig. 4b), and the PERMANOVA results also reported that no significant differences can be found in the rCGC profiles ($P_{rCGC} = 0.128$ with Jaccard index) nor for the root

exudate metabolite uptake profiles ($P_{cat} = 1.00$ with Binomial index, stratified using bacterial isolate labels) between negative and positive rhizosphere response groups. We also trained random forest models predicting the rhizosphere competence for bacterial isolates with either genomic rCGC profiles or with root exudate metabolite uptake profiles (for convenience, hereafter also referred as catabolism-based models and rCGC-based models respectively). For the rCGC-based model, we achieved an accuracy of 0.729 (SD = 0.177), which is lower than the rCGC-based models in the *Pseudomonas* limited cases (Fig. 4c–e). For the catabolism-based model, we only obtained an accuracy of 0.577 (SD = 0.169) (Fig. 4e), and averaging the predictions from repeated measurements did not increase its accuracy (0.542, SD = 0.173, Fig. 4f).

In general, we observed lower prediction accuracies in the second case study, compared to those in the previous one. An obvious explanation is because of the smaller number of (independent) observations available in model training (60 vs. 39 or 16). Our grouped-cross-validation method (more details in *Methods*) accounted for the mutual dependency among repeated measurements in the exometabolite profiling data. Without proper treatment, this could cause overestimation of the prediction accuracy. For example, the prediction accuracy of the catabolism-based model for *Avena barbata* was estimated as high as 95.6% with the built-in out-of-bag (OOB) method (which does not account for sample dependencies) in our study. This also explains why our cross-validated accuracy was not comparable to the accuracy implied by the learning curve of the principal component regression (PCR) model published in the original study of Zhalnina et al., which also did not correct for this.

We evaluated the performance metrics of our models with out-of-sample predictions in cross-validation (Supplementary Table 4). All models showed sensitivity values above 0.8, but only the *Arabidopsis* models achieved specificity values above (or equal to) 0.8. One probable explanation could be that rhizosphere-competent bacteria consist of a narrower group of bacteria compared with the diversity of other groups of bacteria in nature, and our training data may not have sufficiently captured this variation. We also compared the out-of-sample predicted probabilities of each pair of rCGC-based and catabolism-based models (Supplementary Fig. 6, Fig. 4f). Overall, the model predictions from the two models were largely consistent (*Arabidopsis* models: Cor = 0.964, $P < 2.2 \times 10^{-16}$; potato models: Cor = 0.796, P = 1.5 × 10⁻¹⁴; *A. barbata* models: Cor = 0.560, P = 0.012). However, prediction consistency declined in strains with low rhizosphere competence for certain models, specifically, low-colonizing strains in the potato models (Cor = 0.560, P = 0.012) and negative-responding strains in the *A. barbata* models (Cor = 0.304, P = 0.348), which is in line with the lower specificity observed. In addition, we also made predictions with our models on 4 isolates showing a non-significant rhizosphere response which Zhalnina et al. used to test their PCR model (Fig. 4d, gray zone). Although two models in our study made consistent predictions, they were not in line with the PCR model published by Zhalnina et al.

By comparing the rankings of variable importance (Supplementary Fig. 4), we found that for rCGCs with high predictive value in the rCGC-based models, their substrate or related metabolites usually also appear, if available in the catabolism data, on the list of important variables in the corresponding catabolism-based models. These matched pairs include the hemicellulose component xylose, the auxin phenylacetate, the biogenic monoamine phenylethylamine, and the ascorbate degradation downstream organic acid threonate (Supplementary Fig. 4). The xylose-catabolizing rCGC appears in the top 10 important variables in the prediction models for both *A. thaliana* and potato in the first case study and the phenylacetate-catabolizing rCGC appears in the top 10 important variables for all three plants. Xylose was found as one of the major sugars in seed, seedling and root exudates of various plant species, while although phenylacetate is not

abundant in root exudates, its biological precursor phenylalanine is present also as one of the core metabolites in various plant root exudates[78–80]. Our findings suggested that their role in mediating rhizosphere colonization may be universal in many plants. In contrast, metabolites with plant-specific predictive value, such as L-proline in the *A. barbata* study, which was also identified as one of the metabolites with elevated exudation during 6–9 weeks of plant growth[4]. As important osmolytes in both bacteria and plants, L-proline and its catabolic downstream metabolites have been reported to mediate rhizobacteria-dependent drought resistance in several studies[81–83]. But it should be pointed out that high variable importances in random forest models do not always imply a positive correlation, and the size of our case study datasets were also limited to make too general conclusions.

Also, the bacterial strains in these two datasets exhibit a strong phylogenetic bias in rhizosphere competence (Fig. 4a, b, point shapes). Especially, in the second case study, *Bacillota*, *Pseudomonadota* and *Acidobacteriota* spp. strains were exclusively positive or negative rhizosphere responders. This bias makes a simple taxonomy-based prediction also performed well for both datasets (Fig. 4c–e, gray bars). In general, we believe that catabolic gene cluster profiles should not be used as the single source of evidence for predicting rhizosphere competence in general, where other factors such as mobility, biofilm formation, biosynthesis of antibiotics and plant immune response[84–86] also contribute to determining rhizosphere competence, but the data show that these profiles do contain information with significant predictive value.

After all, competence can be predicted with relatively high accuracy based on a very limited number of binary features without any taxonomic information (Fig. 4c–e). Combining rhizoSMASH results with metagenomic and/or metatranscriptomic mapping (facilitated by software such as BiG-MAP software, which can directly take output from any antiSMASH family software as input[87]) in the rhizosphere microbiota when exposed to root exudates with different compositions can be a key next step in unraveling how evolutionary changes in root exudate profiles can lead to shifts in the recruitment of beneficial and non-beneficial microbes by plant hosts, for example during domestication trajectories[88–91].

In conclusion, RhizoSMASH is a bioinformatic software package designed to discover catabolic gene clusters in bacterial genomes that are associated with rhizosphere competence. We studied rCGC presence/absence profiles predicted by the working version of rhizoSMASH across a broad range of bacterial taxonomic groups and found functional diversification at various depths of taxonomic ranks. We also found uneven local distribution and replicon biases of rCGCs throughout genomic sequences in several groups of bacteria. To link rhizoSMASH-detected rCGCs with rhizosphere competence, we performed two case studies, in which we compared the prediction accuracies of rhizosphere competence between rCGC-based and catabolism-based random forest models. Results from these two datasets confirm that the presence/absence profiles of the limited collection of rCGCs carry a comparable amount of niche-specific information compared to the catabolism assays that involve a considerably larger number of metabolites. On the whole, these results indicate an important role of these rCGCs in the adaptation to the rhizosphere niche across the phylogenetic tree.

## Methods

### Construction of the BARS genome collection
In order to facilitate rhizoSMASH analyses of catabolic diversity across bacterial taxa, we compiled the BARS (*Bacteria in the Rhizosphere Soil*) genome assembly collection, containing genome accessions for soil- and rhizosphere-associated bacteria that are available in the NCBI GenBank database. The version until this manuscript has 1226 genome accessions in total. RhizoBase consists of seven subcollections

(REFSOIL, RHIZATHA, SOILATHA, RHIZHVUL, RHIZOSAT, RHIZTAES, RHIZSLYC), with a few overlapping accessions. Among these, the REFSOIL collection contains 842 bacterial entries cited from the RefSoil database[61]; RHIZATHA and SOILATHA came from the at-RSPHERE project which has genome assemblies of 194 Arabidopsis rhizosphere bacteria and 32 bulk soil bacteria[92]; RHIZHVUL, RHIZOSAT, RHIZTAES and RHIZSLYC have 46, 45, 48 and 21 genome accessions respectively, covering bacteria isolated from barley, rice, wheat and tomato rhizosphere. Detailed references and collection sizes are listed in Supplementary Table 1. These genome accession collections and assembly files (GenBank and FASTA files) can be downloaded and locally reconstructed using scripts at https://git.wur.nl/rhizosmash/rhizosmash-case-studies/-/tree/main/RhizoBase.

### rKnownCGC
The rKnownCGC is a database of annotated known rCGCs that are used as reference gene clusters in the KnownClusterBLAST module in rhizoSMASH. The database was manually compiled: known rCGCs were first summarized in YAML formatted text files, which contain information including the genome accession, genomic position, gene function annotations; then a script was used to automatically download source genome sequences, clip the gene cluster region and add extra annotations to the records. A database file for KnownClusterBLAST can be downloaded using the download-rhizosmash-databases command during the installation of rhizoSMASH. Alternatively, a raw database can be reconstructed using scripts at https://git.wur.nl/rhizosmash/rhizosmash-dev/-/tree/main/rKnownCGC. The version used in the analysis of this manuscript contains 92 hand-curated known rCGC sequences and their annotations, comprising 395 protein-coding genes from 56 GenBank records.

### Workflow of rhizoSMASH
RhizoSMASH adopted the biosynthetic gene cluster detection algorithm from antiSMASH version 6[34]. Similar to other antiSMASH variants such as gutSMASH[35], rhizoSMASH takes a bacterial genome sequence as input. If the sequence is not annotated, rhizoSMASH will use prodigal to determine the position of gene open reading frames[93]. A curated set of 271 profile hidden Markov models (pHMMs) is used to screen the genome for functional domains in catabolic pathway enzymes[94]. Then a set of "detection rules", which are logical combinations of pHMMs, is used to identify genome regions that potentially contain any rCGC. Detection rules are divided into 6 categories based on the properties of each pathway's substrate(s), which are carbohydrate, amino acid, organic acid, amine, phytohormone and aromatic compound. By default, rhizoSMASH knits the outcome into an HTML page that the user can interact with and visually investigate. In the version until this manuscript, the user can activate the KnownClusterBLAST module to identify close matches of rhizoSMASH-identified gene clusters to known rCGCs in the rKnownCGC database. The source code and installation guide for rhizoSMASH can be found at https://git.wur.nl/rhizosmash/rhizosmash. The specific working version used to produce results in this manuscript can be found under release tag at https://git.wur.nl/rhizosmash/rhizosmash/-/releases/rhizo-0.3.1. Information of the set of pHMMs used in this version can be found in Supplementary Data 1.

### Creating and Tuning rCGC detection rules
To create detection rules for rCGCs, we underwent three rounds of rule calibration and curation. We first constructed an initial set of detection rules based on known rCGCs in a preliminary literature search. The detection rules contain logical combinations of pHMMs from various sources (193 from Pfam, 64 from TIGRFAM, 4 from NCBIfam, 5 from PANTHER and 5 custom pHMMs). For custom pHMMs, we first used the gene cluster function on the KEGG GENES database to identify genes with conserved genomic contexts, and then

manually selected sequences from distinct clades to build custom pHMMs with the hmmbuild tool of the HMMER software package. Initial thresholding of custom and PANTHER pHMMs was performed with hmmsearch against the Reference Proteomes (RP-35 or PR-55) databases. This set of rules were installed into rhizoSMASH and used to screen RhizoBase genomes. The predicted gene clusters were collected from the output and sent to BiG-SCAPE (a customized version 1 was used the first round; version 2 was used in the second and third rounds) together with known rCGCs. BiG-SCAPE groups gene clusters into cluster families based on their mutual similarities and generates interactive sequence similarity networks. The outputs from BiG-SCAPE were then manually analyzed and verified; we visually investigated if the detected gene clusters were grouped with any known clusters or if they had similar genomic context with any known clusters. Hinted by the outcome, further literature studies were performed to refine the detection rules: removing, adding, splitting or merging them, especially in the first round. Then, the logic operators of the rules and the bit-score thresholds were modified according to the structural characteristics and distinctions of the gene cluster families identified based on analysis of the networks and associated visual alignments. The updated rules were re-installed into rhizoSMASH and underwent another round of verification repeatedly.

The final round of verification used a deduplicated subset of the BARS collections. The mutual similarity of genome sequences in BARS was estimated using mash[95], BARS genomes were first indexed using mash sketch with additional options -k 32 (k-mer size 32); their mutual similarities were then estimated using mash dist. BARS genomes showing mash similarity indexes less than 0.15% were merged, where the most completely assembled genome was kept as representative. Details can be found in the documentation of this repository at https://git.wur.nl/rhizosmash/rhizosmash-case-studies/-/tree/main/gene-cluster-distribution.

### Family-wise distribution of rCGCs
To summarize the taxonomic distribution of rCGCs across BARS genomes, we first selected BARS entries for bacterial families that have sufficient number (≥8) of completely assembled genomes. An rCGC presence/absence table of these genomes in these families was obtained from the rhizoSMASH output in the final rule verification round. Based on the presence/absence table, we calculated in each family the *prevalence* of each rCGC type, the *richness* of all rCGC types, and the *diversity* of rCGCs between genomes. The *prevalence* of an rCGC in a family is defined as the percentage of genomes where the rCGC type is found (Fig. 2a). The rCGC *richness* of a family is represented by the average count of rCGC types present per genome in the family (Fig. 2b). The *diversity* of rCGCs in a family is reflected by the average differences in the presence/absence profiles between each pair of genomes in the family.

### Phylogenetic trees
The phylogenetic tree for bacterial families were built based on lineage information obtained from NCBI with a script using the python module ete3. The lineage information was loaded into R and converted into a data.frame. And the tree was made by applying the as.phylo function in the ape package on that data.frame. The phylogenetic tree for *Nocardiaceae* spp. strains was built according to their 16S rDNA sequences. We first detected and extracted 16S rDNA from the genomic DNA sequences of eight *Nocardiaceae* spp. strains (combined with an outgroup strain *Pseudomonas putida* NBRC 14164) with barrnap[96]. Multiple sequence alignment of these 16S rDNA sequences was performed with MUSCLE[97]. We used FastTree to generate a phylogenetic tree from the multiple sequence alignment[98]. The phylogenetic tree was loaded into R and rooted with the root function in the ape package with the parameter resolve.root set to TRUE. The outgroup tip was removed before final visualization. To make a phylogenetic tree of all

BARS genomes, we utilized GTDB-tk (version 2.4.1[99] with reference data release R226[100]). The genome sequences were first analyzed using the gtdbtk classify_wf workflow. The phylogenetic inference was performed on the multiple alignment of user-provided sequences in the output using the gtdbtk infer command. The tree was rooted with the phylum *Chloroflexota* and visualized using iToL[101].

### Case study datasets
The dataset of 60 phenazine-producing *Pseudomonas* isolates used in case study 1 was published by Zboralski et al. in 2018. All 60 isolates in this dataset have complete genome assemblies available in GenBank. Their catabolic capacity data were measured using Biolog phenotype microarrays. Their rhizosphere colonization strength data were measured in gnotobiotic systems for two plant species (*Arabidopsis thaliana* and potato) using quantitative PCR targeting DNA from a conserved phenazine biosynthesis gene. The results were categorized into "high", "medium" and "low" levels.

The Mediterranean grassland soil bacteria dataset used in case study 2 was published by Zhalnina et al. in 2018. This dataset has 39 sequenced isolates with assembly completeness varying from contig to scaffold levels. Their genome assemblies are available on JGI genome portal under GOLD study ID Gs0017561 or proposal ID 653. The catabolic capacity data in this dataset was obtained with an exometabolomic method: they harvested *Avena barbata* root exudates and compared the percent change of each component before and after bacteria cultivation in the exudates with mass spectrometry. Rhizosphere colonization levels in this dataset were quantified based on enrichment of rhizosphere 16S rDNA in response to the growth of *Avena fatua* during a time course from 0 to 12 weeks. Twenty-seven out of 39 isolates in this dataset showed a significantly positive ($n = 19$) or negative ($n = 8$) rhizosphere response, within which, 12 isolates have catabolism data (2 to 4 replications for each isolate).

### Prediction Models
We used R package randomForest to train prediction models for rhizosphere competence. The caret and rsample packages were used for facilitating cross-validation. In case study 1, we merged "high" and "medium" rhizosphere colonization levels into "med-high" to solve imbalance in the data labeling. In-total, 4 random forests were trained with either rCGC presence/absence data or catabolism data to predict rhizosphere competence levels in *Arabidopsis thaliana* or potato. An 8-fold cross validation was performed for each model to optimize the hyperparameter mtry within a grid. The out-of-sample prediction accuracies for these models were further estimated using an outer layer of 8-fold nested cross-validation. Similar settings were adopted in case study 2 but with different methods for cross-validation due to a smaller data size and different approaches were applied in the rCGC-based and the catabolism-assay-based models. For the rCGC-based model, the hyperparameter tuning was done with the built-in OOB method and a nested 8-fold cross-validation was applied to estimate the prediction accuracy. For the catabolism-assay-based model, to prevent information leakage due to repeated catabolism measurements, the hyperparameter tuning was done by repeating a 2-fold grouped cross-validation (GCV) 4 times, and the prediction accuracy was estimated by repeating a 4-fold GCV 2 times. Data partition for the regular cross-validations was done with the createFolds function in the caret package. For the GCVs, it was done with the group_vfold_cv function in the rsample package, where the strain names and rhizosphere competence labels were set to the group and strata as parameters

We also used a simple taxonomy-based method to predict rhizosphere competence phenotypes. These predictions were made by finding the dominant phenotype in each taxonomy label. We applied 8-fold cross-validation to estimate the accuracy of this method, where

in each training fold, the dominant phenotype in each taxonomy label was recalculated.

## Statistical tests

We used Fisher's exact test to test the complimentary distribution of two trehalose rCGCs across BARS genomes (Fig. 2a). We used the one-sided Wilcoxon Rank Sum test to compare the genomic richness of rCGC types (number of present rCGC types in genomes) between bacterial phyla (Fig. 2a) and environmental and clinical *Nocardiaceae* spp. genomes (Fig. 2b), and the Kruskal-Wallis rank sum test to test if there are differences in the presence of rCGC (count per million bases in genomic sequences) between replicons in *Burkholderia* spp. genomes (Fig. 3b, c). These statistical tests were performed with the fisher.test, the wilcox.test function and kruskal.test function in the stats package in base R. P-values in the *Burkholderia* replicon study was adjusted with the Bonferroni method with the p.adjust function from the stats package in base R. To test whether rCGCs are evenly distributed on chromosomes (Supplementary Fig. 3), we applied a Monto Carlo simulation of 999 repetitions of the mean nearest neighborhood distance (MNN) from a uniform distribution. The $p$ values were estimated by the fraction of repetitions where the simulated MNN was smaller than the observed MNN.

We used the `metaMDS` function from the R package `vegan` to perform NMDS throughout our study. To study the distribution of rCGC type across bacterial taxonomy, the binomial index was used to measure the dissimilarity of rCGC prevalences between phyla (Supplementary Fig. 2). To study the functional biases in replicons of *Burkholderia*, we first labeled the replicons in each genome according to their Jaccard index was used to measure the dissimilarity of rCGC presence/absence profiles in the (Fig. 3b). Similarly, the Jaccard index was used in the NMDS of rCGC profiles (Fig. 4a, b). For the catabolism-assay-derived data, the Jaccard index was used in the first case study and the binomial index was use in the second case study according to their type of data (Fig. 4a, b). PERMANOVAs were conducted with the `adonis2` function also from the `vegan` package and the same dissimilarity indexes were used as in the NMDS analyses. Besides, in the PERMANOVA of metabolite consumption in bacterial strains in case study 2, because of repeated measurements, we added strain labels as `strata` to the parameter to ensure PERMANOVA was performed between strains. The number of permutations was set to 999 as the default value in all the PERMANOVA tests.

## Reporting summary

Further information on research design is available in the Nature Portfolio Reporting Summary linked to this article.

# Data availability

The genome assemblies in the BARS collection and the rKnownCGC database can be reconstructed with the scripts at https://git.wur.nl/rhizosmash/rhizosmash-dev. Datasets for two case studies can be found in the corresponding cited studies, or as formatted copies under corresponding folders at https://git.wur.nl/rhizosmash/rhizosmash-case-studies. The descriptions of rCGCs used in the working version of rhizoSMASH, the description of each BARS genome, and the rCGC presence/absence profiles generated in this study are available in Supplementary Data 1.

# Code availability

Source code for rhizoSMASH is available at https://git.wur.nl/rhizosmash/rhizosmash. A frozen version used to produce results in this manuscript can be found under the release tag at https://git.wur.nl/rhizosmash/rhizosmash/-/releases/rhizo-0.3.1. An archive of this frozen version can also be downloaded from Zendo (https://doi.org/10.5281/zenodo.16780331). Analysis scripts for reproducing the results

and visualization in our manuscript were deposited at https://git.wur.nl/rhizosmash/rhizosmash-case-studies.

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

## Acknowledgements

This work has received funding from The National Key Research and Development Program of China (2021YFD1900200, 2023YFD1902603, 2024YFD1702003), the Chinese Universities Scientific Fund 2024TC063, the Program of Advanced Discipline Construction in Beijing

(Agriculture Green Development), and the 2115 Talent Development Program of China Agricultural University to C.S. Y.L. and M.S. were supported by a project within the Agriculture Green Development (AGD) PhD program of China Agricultural University and Wageningen University & Research (to L.M., C.S., and M.H.M.). The contribution of J.M.R. was funded by the NWO-ENW-funded Gravitation project Microp.

## Author contributions

C.S. and M.H.M. initially conceived the project, with modifications and extensions introduced based on input from J.M.R. and L.M. Y.L. performed data collection from the literature and genomic data analysis, with support from M.H.M. C.S. and M.H.M. coordinated and supervised the overall study, with co-supervision from J.M.R. and L.M. Y.L., M.S., J.M.R., L.M., F.Z., C.S., and M.H.M. contributed to data interpretation. Y.L. prepared the first draft of the manuscript with input from all co-authors. All authors reviewed and approved the final version of the manuscript.

## Competing interests

The authors declare no competing interests.
