## [Transparent Peer Review file · Nature Communications]

Predicting rhizosphere-competence-related catabolic gene clusters in plant-associated bacteria with rhizoSMASH

Corresponding Author: Professor Marnix Medema

Version 0:

Reviewer comments:

Reviewer #1

(Remarks to the Author)

The manuscript "Predicting rhizosphere competence related catabolic gene clusters in plant-associated bacteria with RhizoSMASH" describes a computational workflow to annotate microbial catabolic pathways (rCGC) that underlie rhizosphere competence traits. The workflow entails using profile HMMs (Hidden Markov Models) to detect relevant domains and imposing logical rules that encode presence/absence of combinations of those domains. The rules are chosen based on a literature survey and also "tuned" so that the predictions' sensitivity/specificity is comparable to an independent set of "known rhizosphere adaptation catabolic gene clusters. The tool is then applied to soil isolate genomes to investigate the taxonomic distribution of the detected rCGCs. Two studies where isolates were classified as rhizosphere competent/responding based on either colonization success (using phenazine synthesis gene as a proxy) or abundance pattern during root growth (by mapping 16S amplicon reads to isolate genomes) were used to ask whether those classifications can be recapitulated based on rCGC profiles.

Overall, the manuscript is well motivated, well written, and the details of the work are reasonably well documented (though see below for some suggestions to improve). I think the community working on plant-microbe interactions and generating genome sequence data from isolates or metagenomes will benefit from this tool.

My specific comments and edit suggestions follow below:

- The part of the manuscript that describes the taxonomic distribution of rCGCs collapses genome level information into family level, which I think it is a missed opportunity to quantify the level of phylogenetic conservation of these traits. What was the rationale for this-does that mean these traits were well conserved at family level. Doing a similar analysis at the genome level and then asking how much of the variance can be attributed to taxonomy would be more informative. In addition, one could quantify the phylogenetic conservation of these traits (different aspects of it with different metrics) using a well resolved phylogeny rather than a taxonomic cladogram.

- Figure 2B legend states "abundance" but what is really meant here is "richness". Is that right? There is no abundances involved here as in an estimate of the genome copy number.

- The rules have been "tuned" in an expert informed fashion to get to a reasonable balance of precision and recall on a single curated database. I couldn't find some documentation on what was learned during that tuning. How was precision and recall (sensitivity) was balanced out formally? Was this somewhat ad-hoc with different thresholds for each rule? The work does make a decision on a single dataset-which is ok- but I think it is important to give the reader a sense on the details of what is learned or at least point this out.

-The previous point is also important for genomes of lower quality than isolate genomes (such as MAGs or SAGs). Many users might apply the tool to MAGs, of various completeness and contamination levels, chimeric assemblies etc, so from that perspective the rules that will work "best" under those situations might have to be less "strict" (accepting higher false positive rates).

- Line 150- sentence is not complete

- Line 173- verb tense seems not coherent (suggests should be suggested?)

- Line 247 should refer to Zhahnina et al.

- Consider removing references to figures in the conclusion

- The list of HMMs and rules aren't included in the supplementary materials. The details of where the HMMs come from are lacking in the methods, and one needs to refer to the repository. Please add that to improve readability. Also, the tool will be updated in the future while the manuscript describes a specific instance of the HMM selection and the rules. It is really important to make a frozen version of the databases that were used in the actual manuscript.

- Regarding case studies, the perspective of the authors was to use them as "experimental gold truths" and compare the tool with them in terms of agreement. The small sample sizes (number of isolates) in both studies is an impediment here. While useful to know, that might not make complete sense as rhizosphere competence isn't exactly defined and there is no standard assay for it. "Catabolism-assay" is used to refer to the case study data, which is not correct and is confusing (sounds like there is such an assay. Actually, such studies are valuable from the perspective of discovering novel genomic loci that underlie rhizosphere competence to expand the rulesets that rhizoSMASH has.

Specifically, the first study (Zboralski et al.) used both qPCR and genomic data to classify the isolates, though it is not clear to me how they were weighted as features in the final classification. I wonder how much of the agreement between RhizoSMASH and that specific study is due to the fact that similar features (from genomic data in the study and RhizoSMASH) being used in the classifiers.

The second study, which I am very familiar with as a coauthor, uses differential abundance of the isolates -as they respond to Avena root growth- as a proxy for rhizosphere competence (no genomic features went into the classification). The authors show that a classifier that uses rhizoSmash features doesn't predictively classify the genomes into those groups. I would like to know which, if any, of the individual rules differed between the Zhalnina et al. groups (same for Zboralski but it will be more illuminating for Zhalnina et al). Is it the case that none of them differed or enough of them so random forest classifier's predictions were just random.

The authors might want to add these details about these nuances in comparing with case studies to the discussion and the tool is useful regardless to ask future questions about the genomic basis of rhizosphere competence.

(Remarks on code availability)

Reviewer #2

(Remarks to the Author)

I really enjoyed reading this manuscript by Li et al. "Predicting Rhizosphere Competence Related Catabolic Gene Clusters in plant-associated bacteria with RhizoSMASH". This is coming from the team behind antiSMASH, state of the art method for discovering gene clusters responsible for production of natural products. The authors mined 58 rules for detection of rCGCs, and iteratively improved them using unlabelled data. I believe similar to antiSMASH this tool will be widely used by researchers in microbial and plant biology. The manuscript is very well written. Please see my minor comments below:

- "Specifically, Bacillota genomes contain more carbohydrate-catabolism¹⁴⁴ associated rCGCs (Wilcoxon, $P=0.0219$), consistent with a previous phylogenetic study on the distribution of carbohydrate-activate enzymes (CAZymes) in bacteria⁴³. In contrast, aromatic compound- and phytohormone-catabolizing rCGCs are more frequently found in Pseudomonadota and Actinomycetota (Wilcoxon, $P<2.2\times 10^{-16}$). Indeed, most known bacterial aromatic catabolic pathways^{44,45}, as well as auxin biosynthesis and degradation pathways^{46,47}, were characterized within these two phyla."

I do not think Sup Fig 2 (at least in its current form) provides any useful information and supports any of the claims made by the author. I liked Main text Figure 2 much better, and I think it actually supports the claims made by the authors. However, they need to play with it a little bit, and highlight distinct taxa better.

- The catabolic capacity data in this dataset was obtained with an exometabolomic method: they harvested Avena barbata root exudates and compared the percent change of each component before and after bacteria cultivation in the exudates with mass spectrometry.

If the data before is available, is there any correlation between top molecules (before cultivation) and top pathways that dictate competence in case of Avena Barbata ?

- Figure 4A and B are too crowded, and it is difficult to distinguish between the colors (specially green and yellow)

- I got really confused with evidence levels A/B/C/D in Supp Fig 1 and Supp Table 2 ? I thought they are related to naming of detection rules A/B/C in Figure 1, but apparently they are not. Please put description (genetic / expression / Ecological / Catabolic) instead of A/B/C/D.

- Supp Table 2: Substate -> Substrate

- Names inside Supp Figure 2 (e.g. vanillate) are impossible to see. Text colors have to be bolder.

(Remarks on code availability)

Version 1:

Reviewer comments:

Reviewer #1

(Remarks to the Author)

I went through the authors' rebuttal and point by point responses. I am happy with the current version of the manuscript and agree to its publication in the current version.

(Remarks on code availability)

I went through the authors' rebuttal and point by point responses. I am happy with the current version of the manuscript and agree to its publication in the current version.

Reviewer #2

(Remarks to the Author)

The authors have addresses all my comments. I do not have any further comments, and I believe the manuscript is in an excellent shape now.

(Remarks on code availability)

ANSWER TO REVIEWERS' COMMENTS

Manuscript ID: NCOMMS-25-21325

Title: Predicting rhizosphere-competence-related catabolic gene clusters in plant-associated bacteria with rhizoSMASH

Authors: Yuze Li, Mingxue Sun, Jos M. Raaijmakers, Liesje Mommer, Fusuo Zhang, Chunxu Sone, Marnix H. Medema

Submitted to: Nature Communications

Dear Editor,

We would like to thank you and the reviewers for the constructive comments and valuable suggestions on our manuscript entitled “Predicting rhizosphere-competence-related catabolic gene clusters in plant-associated bacteria with rhizoSMASH”. We have carefully revised our manuscript carefully according to their comments. Below, we provide a detailed point-by-point response to each comment. The reviewer’s comments are reproduced in *italics*, followed by our responses. All changes and new additions have been highlighted with track changes in the revised manuscript. Line numbers in red refer to those in the revised manuscript track changes. Because the tracked changes inflate the line numbering, the line numbers are different in the clean manuscript; these line numbers are indicated by the numbers in parentheses. Quotes of new statements from the revised manuscript will be presented in indented paragraphs.

Reviewer #1

General assessment:

The manuscript “Predicting rhizosphere competence related catabolic gene clusters in plant-associated bacteria with RhizoSMASH” describes a computational workflow to annotate microbial catabolic pathways (rCGC) that underlie rhizosphere competence traits. The workflow entails using profile HMMs (Hidden Markov Models) to detect relevant domains and imposing logical rules that encode presence/absence of combinations of those domains.

The rules are chosen based on a literature survey and also “tuned” so that the predictions’ sensitivity/specificity is comparable to an independent set of “known rhizosphere adaptation catabolic gene clusters. The tool is then applied to soil isolate genomes to investigate the taxonomic distribution of the detected rCGCs. Two studies where isolates were classified as rhizosphere competent/responding based on either colonization success (using phenazine synthesis gene as a proxy) or abundance pattern during root growth (by mapping 16S amplicon reads to isolate genomes) were used to ask whether those classifications can be recapitulated based on rCGC profiles.

Overall, the manuscript is well motivated, well written, and the details of the work are reasonably well documented (though see below for some suggestions to improve). I think the community working on plant-microbe interactions and generating genome sequence data from isolates or metagenomes will benefit from this tool.

Response:

We are glad to read that the reviewer finds our tool and results of interest and the manuscript generally clearly written.

Comment 1:

The part of the manuscript that describes the taxonomic distribution of rCGCs collapses genome level information into family level, which I think it is a missed opportunity to quantify the level of phylogenetic conservation of these traits. What was the rationale for this-does that mean these traits were well conserved at family level. Doing a similar analysis at the genome level and then asking how much of the variance can be attributed to taxonomy would be more informative. In addition, one could quantify the phylogenetic conservation of these traits (different aspects of it with different metrics) using a well resolved phylogeny rather than a taxonomic cladogram

Response:

We agree with the reviewer’s suggestion that is also valuable to show the diversity of rCGCs at the genome level. In response, we added a new supplementary figure (**Supp. Figure 2**) and referred to it in lines **187, 211, 215, 220 and 242** (115, 139, 143, 148 and 157) of the revised manuscript. This figure contains an phylogenetic tree resolved with GTDB-Tk, along with genome assembly completeness, the subcollection labels, and the rCGC profiles of each genome. To facilitate comparison with **Figure 2a**, the corresponding families were marked with gray stripes in **Supp. Figure 2**. This addition provides a more detailed view of the diversity of rCGCs across various taxonomic levels.

Comment 2:

Figure 2B legend states “abundance” but what is really meant here is “richness”. Is that right? There is no abundances involved here as in an estimate of the genome copy number.

Response:

Thank you for pointing out the imprecise use of terminology. We agree that “richness” is more appropriate than “abundance” in this context, considering their definitions in ecology. We have corrected this throughout the manuscript in our revision -- specifically in lines 219, 221, 223 and 241 (147, 149, 151 and 157) of the **Results** section, lines 690 and 692 (430 and 432) in the **Methods** section, as well as in the legend of **Figure 2**.

Comment 3:

The rules have been “tuned” in an expert informed fashion to get to a reasonable balance of precision and recall on a single curated database. I couldn’t find some documentation on what was learned during that tuning. How was precision and recall (sensitivity) was balanced out formally? Was this somewhat ad-hoc with different thresholds for each rule? The work does make a decision on a single dataset-which is ok- but I think it is important to give the reader a sense on the details of what is learned or at least point this out.

Response:

We agree with the reviewer that additional clarification is needed about what was learned during the tuning processes and how the detection rules were modified. In the revised manuscript, we have expanded our explanation of this process in lines 654-660 (397-403) and 66-672 (411-415) as:

The detection rules contain logical combinations of pHMMs from various sources (193 from Pfam, 64 from TIGRFAM, 4 from NCBIfam, 5 from PANTHER and 5 custom pHMMs). For custom pHMMs, we first used the gene cluster function on the KEGG GENES database to identify genes with conserved genomic contexts, and then manually selected sequences from distinct clades to build custom pHMMs with the `hmmbuild` tool of the `hmmer` software package. Initial thresholding of custom and PANTHER pHMMs was performed with `hmmsearch` against the Reference Proteomes (RP-35 or PR-55) databases.

and

Hinted by the outcome, further literature studies were performed to refine the detection rules: removing, adding, splitting or merging them, especially in the first round. Then, the logic operators of the rules and the bit-score thresholds were modified according to the structural characteristics and distinctions of the gene cluster families identified based on analysis of the networks and associated visual alignments.

As described there, the detection rules were iteratively adjusted based on expert knowledge to improve agreement between predicted and known rhizosphere-related gene clusters in a curated dataset. However, it is challenging to quantify precision and recall for all detection rules, as there is no comprehensive ground-truth data with experimental evidence available for the presence and absence of instances of each gene cluster type.

Comment 4:

The previous point is also important for genomes of lower quality than isolate genomes (such as MAGs or SAGs). Many users might apply the tool to MAGs, of various completeness and contamination levels, chimeric assemblies etc, so from that perspective the rules that will work “best” under those situations might have to be less “strict” (accepting higher false positive rates).

Response:

Thank you for your comment concerning the strictness of rCGC detection rules when applying rhizoSMASH to genomes of varying quality, such as MAGs or SAGs. Currently, rhizoSMASH includes a limited set of “relaxed” detection rules for aromatic compound degradation gene clusters. The majority of our detection rules are relatively strict because this version emphasizes primary catabolic pathways, which typically involve fewer genes and exhibit more fixed cluster compositions (unlike larger and compositionally more diverse biosynthetic gene clusters typically analyzed by antiSMASH). We appreciate the reviewer’s point that it is important to adapt and test detection rules for metagenomic studies. We are planning to introduce more flexible detection rules to encompass a broader range of metabolic functions for this in the future. Furthermore, the application of rhizoSMASH in metagenomic sequencing data will be addressed in a dedicated follow-up study.

Comment 5-8:

- *Line 150- sentence is not complete*
- *Line 173- verb tense seems not coherent (suggests should be suggested?)*
- *Line 247 should refer to Zhalnina et al.*
- *Consider removing references to figures in the conclusion*

Response:

Thank you for pointing out these issues in our manuscript. We have addressed them throughout the revised version as follows:

- The incomplete sentence previously at line 150 has been revised and completed; now in line 218 (146).

- The verb “suggusts” at line 173 has been changed to “suggested”; now in line 253 (169).
- The citation at line 247 has been corrected to “Zhalnina et al.”; now in line 366 (249).
- All references to figures have been removed from the text in the conclusions; now in lines 583-595 (334-346).

Comment 9:

The list of HMMs and rules aren't included in the supplementary materials. The details of where the HMMs come from are lacking in the methods, and one needs to refer to the repository. Please add that to improve readability. Also, the tool will be updated in the future while the manuscript describes a specific instance of the HMM selection and the rules. It is really important to make a frozen version of the databases that were used in the actual manuscript.

Response:

We appreciate the reviewer's suggestion to improve clarity about the pHMMs used in rhizoSMASH. In the revised manuscript, we added an excel sheet in the supplementary data listing all pHMMs used in rCGC detection, along with their source repositories and bit-score thresholds, and referred to it in line 650 (393).

Regarding version control, we fully agree with the reviewer's concerns about a frozen version of rhizoSMASH at the time point of the manuscript. To address this, we now mention the tagged release of the working version of rhizoSMASH in our repository in the revised manuscript in lines 647-650 (390-392).

Comment 10:

Regarding case studies, the perspective of the authors was to use them as “experimental gold truths” and compare the tool with them in terms of agreement. The small sample sizes (number of isolates) in both studies is an impediment here. While useful to know, that might not make complete sense as rhizosphere competence isn't exactly defined and there is no standard assay for it. “Catabolism-assay” is used to refer to the case study data, which is not correct and is confusing (sounds like there is such an assay. Actually, such studies are valuable from the perspective of discovering novel genomic loci that underlie rhizosphere competence to expand the rulesets that rhizoSMASH has.

Specifically, the first study (Zboralski et al.) used both qPCR and genomic data to classify the isolates, tough it is not clear to me how they were weighted as features in the final classification. I wonder how much of the agreement between RhizoSMASH and that specific

study is due to the fact that similar features (from genomic data in the study and RhizoSMASH) being used in the classifiers.

The second study, which I am very familiar with as a coauthor, uses differential abundance of the isolates -as they respond to Avena root growth- as a proxy for rhizosphere competence (no genomic features went into the classification). The authors show that a classifier that uses rhizoSmash features doesn't predictively classify the genomes into those groups. I would like to know which, if any, of the individual rules differed between the Zhalnina et al. groups (same for Zboralski but it will be more illuminating for Zhalnina et al). Is it the case that none of them differed or enough of them so random forest classifier's predictions were just random.

The authors might want to add these details about these nuances in comparing with case studies to the discussion and the tool is useful regardless to ask future questions about the genomic basis of rhizosphere competence.

Response:

Thank you for this thoughtful and detailed feedback on our case studies. We agree that more precise descriptions on the datasets and further interpretation of the prediction models are needed.

We acknowledge that the use of “catabolism assay” could cause confusion about the methods used the two cited studies. To address this, we have introduced the precise name of each methods at their first mention in the revised manuscript in lines 323-324 (227-228) and in lines 368-369 (251-252), along with references to detailed descriptions in the **Methods** section. After which, we used “catabolic profile” for the Biolog phenotyping data and “root exudate uptake profile” for the exometabolite profiling data to describing the data themselves. Yet, for the names of the random forest models, to avoid long compound words that hinder readability, we used “catabolism-based model” in the remaining text and figure legends.

Regarding the first case study (Zboralski et al.), we revisited their publication and confirmed that the classification of rhizosphere colonization was based solely on qPCR targeting a conserved region of a single gene, which was also mentioned in the **Methods** section now in lines 717-719 (456-459):

Their rhizosphere colonization strength data were measured in gnotobiotic systems for two plant species (*Arabidopsis thaliana* and potato) using quantitative PCR targeting DNA from a conserved phenazine biosynthesis gene.

Although they presented their findings with genome mining along with experimental results, they did not perform machine learning with these results. Hence, there were no classifiers presented in that study using features overlapping with the detected rhizoSMASH-detected gene clusters.

As for the second case study, we sincerely appreciate the reviewer's perspective as a coauthor of the original work which we cited for the dataset. With this dataset, we have trained two random forest classifier models; one was based on rCGCs detected from genome assemblies of their isolates (rCGC-based model), and the other was based on the root exudate uptake data (measured with exometabolite profiling) published in the reviewer's original study (catabolism-based model). In our results, the rCGC-based model demonstrated higher prediction accuracy than the catabolism-based model. We have also noticed the principal component regression model described in the original study that the reviewer was involved in, which showed greater accuracy considering the reported mean squared errors (during metaparameter tuning, Supplementary Figure 13a and 14a in their original study). In the previous version of our manuscript, we did not discuss about the results of this model for two main reasons: (1) we did not find information in their publication supporting their cross-validation method having compensated for the effect of dependent observations in their exometabolite profiling data (repeated measurements for each isolates); (2) the test dataset was not labeled (as these isolates are "n.s." for their response to rhizosphere growth), which means the prediction on this test set cannot be used to generate an estimate of the out-of-sample accuracy. In the revised manuscript of our study, we have now addressed this point by further discussing the comparisons of model performance in lines 386-394 (269-277):

Our grouped-cross-validation method (more details in **Methods**) accounted for the mutual dependency among repeated measurements in the exometabolite profiling data. Without proper treatment, this could cause overestimation of the prediction accuracy. For example, the prediction accuracy of the catabolism-based model for *Avena barbata* was estimated as high as 95.6% with the built-in out-of-bag method (which does not account for sample dependencies) in our study. This also explains why our cross-validated accuracy was not comparable to the accuracy implied by the learning curve of the principal component regression (PCR) model published in the original study of Zhalnina et al., which also did not correct for this.

To provide more detailed evaluation of our models, we added **Supp. Table 4** to report their performance metrics (sensitivity, specificity and precision) and discuss our understanding of these in lines 395-399 (278-283):

We evaluated the performance metrics of our models with out-of-sample predictions in cross-validation (**Supp. Table 4**). All models showed sensitivity values above 0.8, but only the *Arabidopsis* models achieved specificity values above (or equal to) 0.8. One probable explanation could be that rhizosphere-competent bacteria consist of a narrower group of bacteria compared with the diversity of other groups of bacteria in nature, and our training data may not have sufficiently captured this variation.

Additionally, please see also added **Figure 4f** and **Supp. Figure 6** comparing the predicted outcomes of different models targeting the same reference data, and discussed the findings in lines 501-512 (283-294):

We also compared the out-of-sample predicted probabilities of each pair of rCGC-based and catabolism-based models (**Supp. Figure 6, Figure 4f**). Overall, the model predictions from the two models were largely consistent (*Arabidopsis* models: Cor=0.964, $P < 2.2 \times 10^{-16}$; potato models: Cor=0.796, $P = 1.5 \times 10^{-14}$; *A. barbata* models: Cor=0.560, $P = 0.012$). However, prediction consistency declined in strains with low rhizosphere competence for certain models, specifically, low-colonizing strains in the potato models (Cor=0.560, $P = 0.012$) and negative-responding strains in the *A. barbata* models (Cor=0.304, $P = 0.348$), which is in line with the lower specificity observed. In addition, we also made predictions with our models on 4 isolates showing a non-significant rhizosphere response which Zhalnina et al. used to test their PCR model (**Figure 4d**, gray zone). Although two models in our study made consistent predictions, they were not in line with the PCR model published by Zhalnina et al.

We hope these additions adequately address the reviewer's concerns and provide additional insights in how our results compare to previously published work.

Reviewer #2

General assessment:

I really enjoyed reading this manuscript by Li et al. "Predicting Rhizosphere Competence Related Catabolic Gene Clusters in plant-associated bacteria with RhizoSMASH". This is coming from the team behind antiSMASH, state of the art method for discovering gene clusters responsible for production of natural products. The authors mined 58 rules for detection of rCGCs, and iteratively improved them using unlabelled data. I believe similar to antiSMASH this tool will be widely used by researchers in microbial and plant biology. The manuscript is very well written.

Response:

We thank the reviewer for the positive feedback and are glad to read that they think the tool will likely be widely used by the scientific community.

Comment 1:

"Specifically, Bacillota genomes contain more carbohydrate-catabolism¹⁴⁴ associated rCGCs (Wilcoxon, $P=0.0219$), consistent with a previous phylogenetic study on the distribution of carbohydrate-activate enzymes (CAZymes) in bacteria⁴³. In contrast, aromatic compound- and phytohormone-catabolizing rCGCs are more frequently found in Pseudomonadota and Actinomycetota (Wilcoxon, $P<2.2\times 10^{-16}$). Indeed, most known bacterial aromatic catabolic pathways^{44,45}, as well as auxin biosynthesis and degradation pathways^{46,47}, were characterized within these two phyla."

I do not think Sup Fig 2 (at least in its current form) provides any useful information and supports any of the claims made by the author. I liked Main text Figure 2 much better, and I think it actually supports the claims made by the authors. However, they need to play with it a little bit, and highlight distinct taxa better.

Response:

Thank you for your comments about the use of referrals to figures. We agree with the reviewer that the visual clues in **Supp. Figure 2** (now **Supp. Figure 3**) were directly related to the claims cited by the reviewers. We addressed this issue in lines **211 and 215** (139 and 143) by referring to **Figure 2a** and the new **Supp. Figure 2** (genome level rCGC profile with phylogenetic tree) instead.

Comment 2:

The catabolic capacity data in this dataset was obtained with an exometabolomic method: they harvested Avena barbata root exudates and compared the percent change of each component before and after bacteria cultivation in the exudates with mass spectrometry.

If the data before is available, is there any correlation between top molecules (before cultivation) and top pathways that dictate competence in case of Avena Barbata ?

Response:

We appreciate the reviewer's idea of comparing important prediction variables with metabolomic profiles of root exudates. In our revised manuscript, we expanded the discussion of this idea in lines **518-529** (300-311). Especially, we found L-proline, as a metabolite with increased exudation during 6-9 weeks in the study of Zhalnina at al., showed high importance in model prediction. We now discussed the potential functional role of L-proline and contextualize this with literature in these lines:

The xylose- and phenylacetate-catabolizing rCGCs both appear in the top 10 important variables in prediction models for rhizosphere competence in all three plants. Xylose was found as one of the major sugars in seed, seedling and root exudates of various plant species, while, although phenylacetate is not abundant in root exudates, its biological precursor phenylalanine is present also as one of the core metabolites in various plant root exudates⁷⁸⁻⁸⁰. Our findings suggested that their role in mediating rhizosphere colonization may be universal in many plants. In contrast, metabolites with plant-specific predictive value, such as L-proline in the *A. barbata* study, which was also identified as one of the metabolites with elevated exudation during 6 to 9 weeks of plant growth⁴. As important osmolytes in both bacteria and plants, L-proline and its catabolic downstream metabolites have been reported to mediate rhizobacteria-dependent drought resistance in several studies⁸¹⁻⁸³.

Comment 3:

Figure 4A and B are too crowded, and it is difficult to distinguish between the colors (specially green and yellow)

Response:

Thank you for pointing this out. In the revised manuscript, we have enlarged the NMDS plots and applied more distinct, colorblind-friendly color schemes to improve clarity and accessibility in **Figure 4**.

Comment 4:

I got really confused with evidence levels A/B/C/D in Supp Fig 1 and Supp Table 2 ? I thought they are related to naming of detection rules A/B/C in Figure 1, but apparently they are not. Please put description (genetic / expression / Ecological / Catabolic) instead of A/B/C/D.

Response:

Thank you for this suggestion. We agree with the reviewer that the labeling on the diagram was misleading. We have changed the labeling according to the reviewer's suggestion in the revised manuscript.

Comment 5:

Supp Table 2: Substate -> Substrate

Response:

Thank you for pointing out this typo. We have corrected it in the legend of **Supp. Table 2** in the revised manuscript.

Comment 6:

Names inside Supp Figure 2 (e.g. vanillate) are impossible to see. Text colors have to be bolder.

Response:

We have addressed this issue in the revised manuscript by using bolder colors and a larger canvas in the updated figure (now **Supp. Figure 3**).